# Ultrafast infrared nano-imaging of far-from-equilibrium carrier and vibrational dynamics

Jun Nishida [1], Samuel C. Johnson [1], Peter T. S. Chang[1], Dylan M. Wharton [1], Sven A. Dönges [1], Omar Khatib [1] & Markus B. Raschke [1✉]

Ultrafast infrared nano-imaging has demonstrated access to ultrafast carrier dynamics on the nanoscale in semiconductor, correlated-electron, or polaritonic materials. However, mostly limited to short-lived transient states, the contrast obtained has remained insufficient to probe important long-lived excitations, which arise from many-body interactions induced by strong perturbation among carriers, lattice phonons, or molecular vibrations. Here, we demonstrate ultrafast infrared nano-imaging based on excitation modulation and sideband detection to characterize electron and vibration dynamics with nano- to micro-second life-times. As an exemplary application to quantum materials, in phase-resolved ultrafast nano-imaging of the photoinduced insulator-to-metal transition in vanadium dioxide, a distinct transient nano-domain behavior is quantified. In another application to lead halide perovskites, transient vibrational nano-FTIR spatially resolves the excited-state polaron-cation coupling underlying the photovoltaic response. These examples show how heterodyne pump-probe nano-spectroscopy with low-repetition excitation extends ultrafast infrared nano-imaging to probe elementary processes in quantum and molecular materials in space and time.

[1] Department of Physics and JILA, University of Colorado, Boulder, CO 80309, USA. ✉email: markus.raschke@colorado.edu

Functional materials offer intriguing applications based on their unique optical and electronic properties, such as quantum phase transition based Mott transistors[1], polaronic carrier transport in lead halide perovskite photovoltaics[2], coherent phonons and vibrations driving singlet fission[3], or electronic energy transfer in light-harvesting complexes[4]. These properties emerge from the interplay of the elementary electronic, vibrational, and phononic quantum states. By exciting one of these degrees of freedom out of equilibrium and probing their dynamic response, ultrafast spectroscopy disentangles mode-coupling and competing interactions that are otherwise convoluted in static spectroscopy[5]. In addition, when the system is perturbed far-from-equilibrium, it can be driven into new photoinduced quantum states, enabling ultrafast optical control of metallic, superconductive, and polaronic functionalities[6]. Further, strong and ultra-short laser fields lead to extreme nonlinear optical phenomena with applications from high harmonic generation[7,8] to light-field petahertz electronics[9].

However, these ultrafast processes and material functions often exhibit spatial heterogeneities associated with, e.g., lattice defects, strain, grain boundaries, and nonuniform doping from atomic to device scales[10–12]. To address the associated spatiotemporal dynamics, a variety of ultrafast nanoimaging techniques have been developed, including ultrafast transmission electron microscopy (TEM)[13], photoemission electron microscopy (PEEM)[14], or X-ray microscopy[15]. Yet, these techniques often cannot readily resolve the low-energy electronic and lattice coupling and dynamics that critically control the material properties.

In contrast to these established techniques, scattering scanning near-field optical microscopy (s-SNOM) has recently been implemented with ultrafast time resolution[16–28]. In particular, ultrafast infrared nanoimaging based on nano-FTIR spectroscopy[19] and electro-optic sampling (EOS)[24,29] has enabled access to nanoscale carrier dynamics in semiconductors[19,21,24], correlated electron[17], and polaritonic materials[20,23] with spatiotemporal-spectral resolutions. Ultrafast infrared nanoimaging to date has probed strong and short-lived carrier and collective polaritonic excitations based primarily on high-repetition-rate (>10 MHz) laser pump sources, which offer a high duty cycle to enhance the signal-to-noise ratio.

Important long-lived transients in materials arise from many-body interactions induced by a strong perturbation, such as photoinduced phases in correlated electron systems[30–33] or polaron dynamics in organic–inorganic hybrid photovoltaics[34,35]. In these systems, cooperative dynamics among the many degrees of freedom[36,37] result in the formation of nonequilibrium states with nano- to microsecond lifetimes. However, high-repetition excitation precludes quantitative probing of the ultrafast dynamics of those states. Full spatiotemporal-spectral resolution with low-repetition excitation has only been achieved recently[21,23], while the reduced duty cycle limits signal intensity and contrast. This calls for a generalized approach with enhanced excited-state contrast at low-repetition-rate excitation.

Here, we demonstrate non-degenerate heterodyne pump-probe infrared scattering scanning near-field optical microscopy (HPP IR s-SNOM) with low-repetition-rate modulated excitation, which provides simultaneous space, time, frequency, and phase resolutions with high sensitivity. A modulated optical pump excites the system into an excited state, followed by infrared heterodyne probing of the transient low-energy electronic and vibrational response. The induced third-order nano-localized polarization is isolated by sideband lock-in detection and directly detected interferometrically in the time domain, enabling ultrafast nanoimaging with high contrast even with the low-repetition excitation rate of 1 MHz. The excitation-modulated HPP IR s-SNOM thus provides the analog of ground-state nano-FTIR spectroscopy, resolving the transient and nano-localized excited-state response.

The spectrally resolved near-field pump-probe signal is modeled to quantitatively extract the spatiotemporal evolution of the transient dielectric function of the material based on a combination of finite dipole and four-layer reflection model.

As a representative application to quantum materials, we perform nanoimaging of the electron dynamics associated with the ultrafast photoinduced insulator-to-metal transition (IMT) in vanadium dioxide (VO$_2$), resolving transient domain dynamics that is distinct from the established strain-induced heterogeneity in the thermally induced transition[10,16,17,38]. In another application to soft molecular materials, based on transient nano-spectroscopy with the spectral resolution, we directly resolve heterogeneity in polaron–cation coupling that controls the photovoltaic response by probing the vibrational dynamics in a triple cation perovskite[34,35].

The low-repetition-rate excitation with highly sensitive detection leads to the study of photoinduced phase transitions as well as soft photovoltaic materials with long-lived carrier and vibrational responses. The approach thus holds promise to establish the missing links between the elementary processes at the nanoscale and the associated macroscopic optical, photophysical, catalytic, or electronic properties of a wide range of functional materials.

## Results

Figure 1a–c shows the schematics of HPP IR s-SNOM with femtosecond pump (Yb:KGW amplified laser with ~185 fs full-width-at-half-maximum (FWHM) pulse duration, 1030 nm center wavelength, ~1 MHz repetition rate, Pharos, Light Conversion), and broadband infrared nano-probe spectroscopy (tunable at 5–10 μm, ~170 fs FWHM pulse duration, Orpheus OPA/DFG, Light Conversion), with overall time resolution of ~200 fs (see Supplementary Fig. S1), and ~40 nm spatial resolution as given by the apex radius of the metallic scanning probe tip (Supplementary Fig. S4). IR s-SNOM is implemented with an asymmetric Michelson interferometer, consisting of sample and reference arms as established[39,40]. An atomic force microscope (Innova AFM, Bruker) in the sample arm is operated in tapping mode with a tip-tapping frequency $\omega_t$ (for details, see Supplementary Note 1). Pump and probe pulses, separated by time delay $T$, are collinearly directed onto the AFM tip (ARROW-NCPt, NanoAndMore USA) with an off-axis parabolic mirror (NA = 0.45). The fundamental or frequency-doubled pump source is modulated at frequency $\Omega_M$ by an acousto-optic modulator (AOM), or a mechanical chopper. The pump-induced excited-state population and corresponding ground-state bleach (Fig. 1b) are detected by pump-probe s-SNOM spectroscopy. Here, the tip-scattered near-field probe signal $E_{NF}$ is then optically heterodyned with the local oscillator field $E_{LO}$ from the reference arm, with variable delay $t$, and detected by a HgCdTe (MCT) detector. As in conventional IR s-SNOM, lock-in demodulation at $n\omega_t$ ($n = 1, 2, 3...$), combined with the interferometric heterodyne detection, isolates $I_{NF,Het} \sim E_{NF}E_{LO}^*$ that provides background-free nano-localized imaging contrast[39,40].

In contrast to previous pump-probe IR s-SNOM implementations with un-modulated pump excitation[19,24], with excitation modulation at $\Omega_M$ we perform sideband lock-in demodulation at $n\omega_t \pm \Omega_M$ to directly detect the photoinduced change in the near-field signal $\Delta I_{NF}(T)$, similar to a recent implementation in ultrafast THz-EOS s-SNOM[29]. This detection scheme enhances the signal-to-noise ratio by a factor of >4 in comparison to the conventional scheme, corresponding to a reduction in the data acquisition time by more than one order of magnitude (Supplementary Fig. S9). By scanning $E_{LO}$ in time $t$, we then acquire the heterodyne pump-probe interferogram $\Delta I_{HPP}(t, T)$ in the time domain.

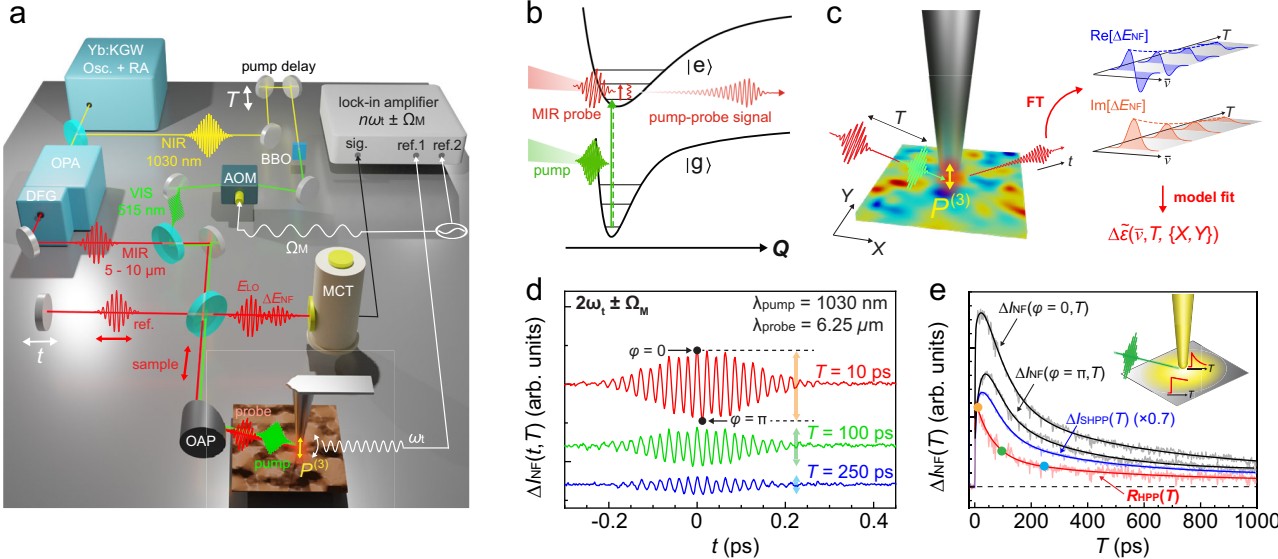

**Fig. 1 Heterodyne pump-probe nanoimaging. a** HPP IR *s*-SNOM, with the ~ 185 fs optical 1.2/2.4 eV (1030/515 nm) pump excitation, ~170 fs tunable mid-IR 0.12–0.25 eV (5–10 μm) probe, and interferometric heterodyne near-field detection. AOM acousto-optic modulator, DFG difference-frequency generation, OAP off-axis parabolic mirror, OPA optical parametric amplifier, Osc.+RA oscillator and regenerative amplifier laser system. **b** Far-from-equilibrium excitation followed by mid-infrared probe of the transient low-energy vibrational and electronic response coupled to the excited state. **c** The tip-localized time domain signal of $\Delta E_{NF}(\overline{\nu})$, from which the pump-induced change in the nano-localized complex dielectric function $\Delta\tilde{\epsilon}_{NF}(\overline{\nu})$ is retrieved with spatiotemporal-spectral resolution. **d** Sideband-demodulated pump-probe interferogram $\Delta I_{NF}(t, T)$ for a Ge reference sample. **e** *T*-dependent pump-probe transients from two different $E_{LO}$ phase values $\Delta I_{NF}(\phi = 0, T)$ and $\Delta I_{NF}(\phi = \pi, T)$, the derived frequency-averaged heterodyne pump-probe amplitude relaxation $R_{HPP}(T)$, and the self-homodyne pump-probe signal relaxation $\Delta I_{SHPP}(T)$. Inset: tip-enhanced pump excitation in nano-localized probe volume, leading to faster relaxation from higher excited carrier density.

$\Delta I_{HPP}(t, T)$ is then Fourier transformed with respect to *t* and deconvolved with $E_{LO}(\overline{\nu})$ to yield the real and imaginary parts of $\Delta E_{NF}(\overline{\nu}, T)$ (Fig. 1c), and with much higher signal-to-noise ratio compared to the pump un-modulated case (see Supplementary Fig. S9 for examples). $\Delta E_{NF}(\overline{\nu}, T)$ quantitatively relates to the transient photoinduced change in the complex dielectric function, which can be retrieved through the model fitting. HPP IR *s*-SNOM thus provides the full 4D characterization of the transient material response with spatial (*X*, *Y*), temporal (*T*), and spectral ($\overline{\nu}$) resolutions, with high excited-state contrast enabled by the selective detection of the pump-induced near-field response.

As an example, Fig. 1d shows a representative pump-probe interferogram $\Delta I_{HPP}(t, T)$ of a germanium reference sample, acquired by continuously scanning the $E_{LO}$ delay *t* at selected pump-probe time delays *T*. The *t*-dependent profiles of $\Delta I_{HPP}(t, T)$ is determined and limited by the spectral profile of the infrared-probe pulse due to the nearly instantaneous and spectrally broad transient free carrier Drude response of germanium[41]. To determine the spectrally averaged amplitude relaxation $R_{HPP}(T)$, $\Delta I_{NF}$ is recorded as *T* is scanned for sets of two distinct $E_{LO}$ phase values, $\phi = 0$ and $\pi$ (see Fig. 1d) for constructive and destructive interference near zero-path difference (ZPD), to yield the amplitude decay of the HPP signal as $R_{HPP}(T) = \Delta I_{NF}(\phi = 0, T) - \Delta I_{NF}(\phi = \pi, T)$ (Fig. 1e). This approach of extracting the spectrally averaged amplitude based on the two-phase measurement is applicable due to the narrower laser bandwidth of ~100 cm$^{-1}$ FWHM compared to the much broader Drude response in germanium (see Supplementary Note 1).

The HPP signal relaxation $R_{HPP}(T)$ (red) probing the recombination of photoinduced carriers is notably distinct from the self-homodyne pump-probe (SHPP) decay $\Delta I_{SHPP}(T)$ (blue), recorded without interference with $E_{LO}$. $\Delta I_{SHPP}(T)$ shows a small initial rise followed by a slow decay[41], while $R_{HPP}(T)$ only shows a decay yet faster than $\Delta I_{SHPP}(T)$. We attribute this difference to the convolution of the time-dependent far-field background in

$\Delta I_{SHPP}(T)$[22]. The tip-enhanced pump field results in higher local carrier density compared to the far-field pumped background (Fig. 1e, inset). With its pure local probe character, $R_{HPP}(T)$ thus exhibits a faster relaxation, reflecting the enhanced recombination and scattering induced by the higher excited carrier density. This observation critically highlights the necessity of interferometric heterodyne detection to quantify the nano-localized pump-probe dynamics, particularly in the high fluence regime where the observed dynamics are highly sensitive to the local pump intensity.

**Theory of HPP IR *s*-SNOM.** To evaluate excited-state absorption resonances at the nanoscale[18,21,23,24], which contain critical information associated with many-body interactions, we extend the dipole model in combination with multilayer reflection, established for both ground-state[42–44] and ultrafast *s*-SNOM[18,24], to quantitatively relate the experimentally observed sideband-demodulated pump-probe interferogram $\Delta I_{HPP}(t)$ to the underlying transient dielectric function $\Delta\tilde{\epsilon}_{NF}(\overline{\nu})$. Importantly, we find that $\Delta E_{NF}(\overline{\nu})$ as the direct observable in transient vibrational nano-spectroscopy is generally the convolution of the excited-state absorption and Fano-type interference. This underscores the importance of the theoretical framework to retrieve $\Delta\tilde{\epsilon}_{NF}(\overline{\nu})$ that purely encodes the excited-state absorption to distinguish these two contributions.

We illustrate the application for the case of a transient molecular vibrational response coupled to photoinduced carriers[35,45], with a 600-nm thick sample film coated on a substrate (Fig. 2a). We assume ground-state and excited-state dielectric functions, $\tilde{\epsilon}^{(0)}(\overline{\nu})$ and $\Delta\tilde{\epsilon}_{NF}(\overline{\nu})$, with vibrational resonances centered at $\overline{\nu}_{gs}$ and $\overline{\nu}_{ex}$, respectively (Fig. 2b). The pump excitation modifies the dielectric function of the sample film to $\tilde{\epsilon}^{(e)} = \tilde{\epsilon}^{(0)} + \Delta\tilde{\epsilon}_{NF}$, down to a depth $d_1 = 100$ nm determined by the absorption coefficient. The remaining depth of $d_2 = 500$ nm is left unperturbed at $\tilde{\epsilon}^{(0)}$.

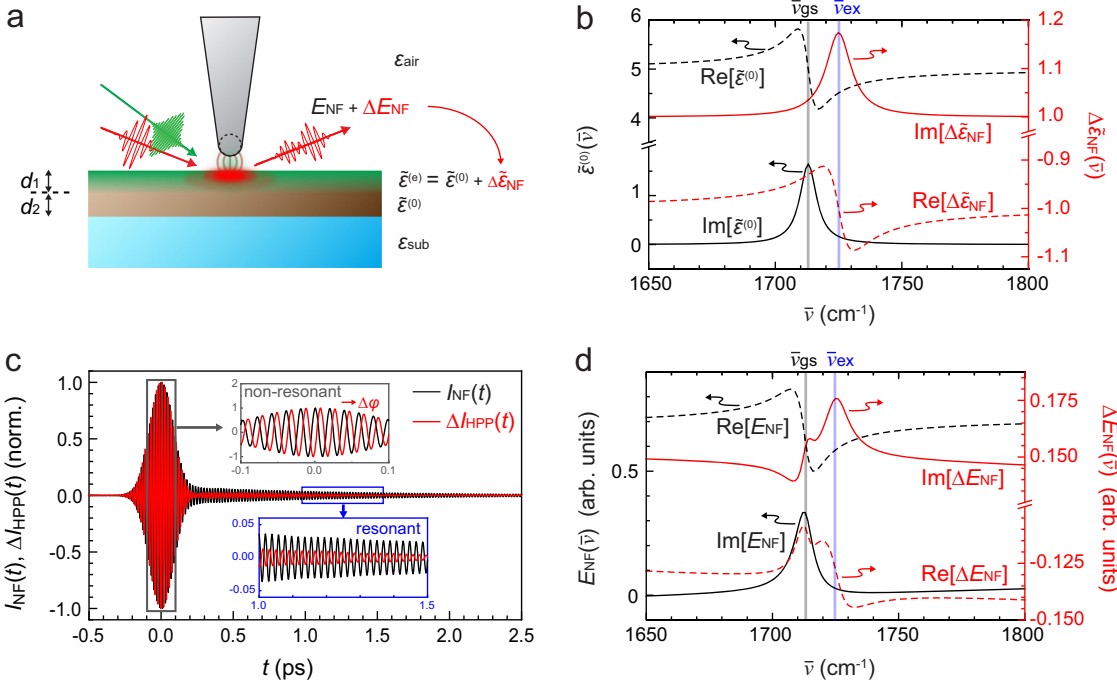

**Fig. 2 Retrieval of the transient nanoscale response function. a** Theoretical framework to quantitatively relate the measured pump-induced near-field signal change $\Delta E_{NF}$ to the transient complex dielectric function $\Delta\tilde{\epsilon}_{NF}$. **b** Example ground-state complex dielectric function $\tilde{\epsilon}^{(0)}(\bar{\nu})$ and its pump-induced change $\Delta\tilde{\epsilon}_{NF}(\bar{\nu})$ for a model vibrational response, with the ground-state resonance at $\bar{\nu}_{gs}$ and the frequency-shifted excited-state resonance at $\bar{\nu}_{ex}$. **c** Simulated heterodyne IR s-SNOM interferograms for the ground-state $I_{NF}(t)$ and its pump-induced change $\Delta I_{HPP}(t)$. **d** $E_{NF}(\bar{\nu})$ and $\Delta E_{NF}(\bar{\nu})$ in frequency-domain, obtained from Fourier transform and $E_{LO}(\bar{\nu})$ deconvolution of the time-domain interferograms in panel **c**.

The near-field scatter with and without pump excitation is calculated based on a combination of finite dipole[42] and four-layer models for a sinusoidally modulated tip-sample distance and is demodulated with the second harmonic tip-tapping frequency. While the finite dipole model is known to quantitatively retrieve vibrational resonances[42], the four-layer reflection model[46] accounts for partial excitation of a material layer. For details, see Supplementary Note 2.

Figure 2c shows the calculated ground-state s-SNOM interferogram $I_{NF}(t)$ (black). The pump-induced response $\Delta\tilde{\epsilon}_{NF}(\bar{\nu})$, which consists of a broadband carrier response and a vibrational excited-state absorption resonance, then gives rise to the pump-probe interferogram $\Delta I_{HPP}(t)$ (red). As expected, the non-resonant term in $\tilde{\epsilon}^{(0)}(\bar{\nu})$ leads to a nearly instantaneous center burst of $I_{NF}(t)$ limited by the duration of the probe pulse followed by a vibrational free-induction decay (FID)[40]. $\Delta I_{HPP}(t)$ is phase-shifted due to the complex broadband response of photoinduced carriers, with the FID from the frequency-shifted photoinduced vibrational resonance.

The Fourier transforms of $I_{NF}(t)$ and $\Delta I_{HPP}(t)$, after deconvolution with $E_{LO}(\bar{\nu})$, yield the spectral profiles of the ground-state $E_{NF}(\bar{\nu})$ (black) and its photoinduced change $\Delta E_{NF}(\bar{\nu})$ (red) as shown in Fig. 2d. As can be seen, $E_{NF}(\bar{\nu})$ approximates the ground-state complex dielectric function $\tilde{\epsilon}^{(0)}(\bar{\nu})$ as established[40,42]. In contrast, the spectral profile of $\Delta E_{NF}(\bar{\nu})$ is a complex convolution of ground- and excited-state dielectric responses. While $\Delta E_{NF}(\bar{\nu})$ exhibits a resonance corresponding to the excited-state absorption at $\bar{\nu}_{ex}$, it is also compounded by Fano-type interference of the vibrational response with the pump-induced broadband carrier response. In the analysis of the experimental data presented below, with the application of the finite dipole model[42] based on Drude-carrier and Lorentzian-vibrational material responses, we retrieve the transient complex dielectric function $\Delta\tilde{\epsilon}_{NF}(\bar{\nu})$ quantitatively from $\Delta E_{NF}(\bar{\nu})$. $\Delta\tilde{\epsilon}_{NF}(\bar{\nu})$ purely

describes the excited-state absorption and thus provides the fundamental electronic and vibrational coupling and their dynamics (see Supplementary Note 2).

We note that Fano-type interference is generally expected to give rise to a nontrivial lineshape in $\Delta E_{NF}$ whenever spectrally distinct and narrow resonances interfere with pump-induced broadband response, as is generally the case for visible-to-UV pump and infrared-to-THz probe nano-spectroscopy of a wide range of molecular and soft materials. The model is generalizable to other types of resonant excitation and is universally applicable to retrieve the underlying ground- and excited-state dielectric functions in HPP s-SNOM, enabling the quantitative separation of excited-state absorption and Fano-type interference.

**Phase-controlled imaging of quantum IMT in vanadium dioxide nanobeam.** We first apply HPP IR s-SNOM to nanoimaging heterogeneity in the photoinduced quantum phase transition and associated electron dynamics in a nanobeam of the correlated electron material vanadium dioxide ($VO_2$). $VO_2$ exhibits a thermal insulator-to-metal transition (IMT) at ~340 K, which involves a bandgap collapse with a change from an insulating monoclinic to a metallic rutile phase[10,38,47,48]. The IMT can also be induced optically[10,16,17,30,32,33] with promising applications for, e.g., ultrafast photoswitches[48].

The phase transition is believed to be caused by a complex interplay between electron–electron correlation and electron–phonon coupling, with details regarding the exact mechanism still remaining elusive[49,50]. As a manifestation of its intricacy, both the thermally induced and photoinduced transition of $VO_2$ exhibit spatial heterogeneity and are susceptible to local strain and chemical heterogeneity[10,16,17,38,47]. Recent optical pump/infrared-probe s-SNOM of the photoinduced IMT of $VO_2$ revealed a spatial profile distinct from that of thermal strain-induced heterogeneity and was attributed to possible stoichiometric zoning[16]. However,

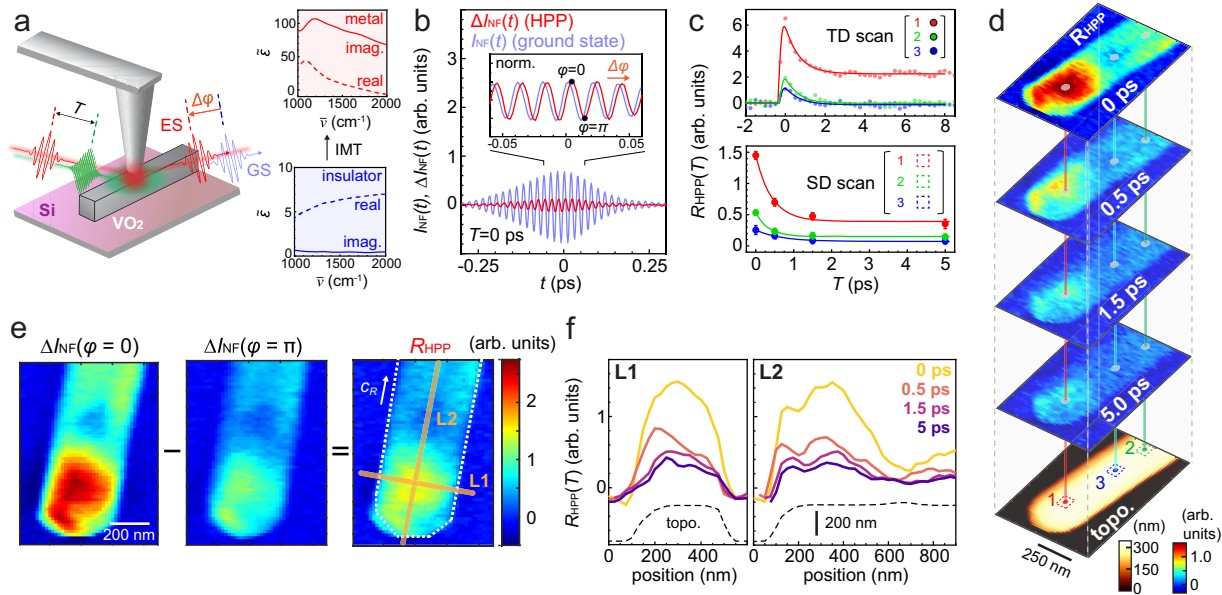

**Fig. 3 Phase-controlled ultrafast nanoimaging of insulator-to-metal transition in a VO₂ nanobeam. a** Schematic of NIR pump (1.2 eV, ~2 mJ/cm²) infrared-probe (0.2 eV) ultrafast nanoimaging of VO₂ (left) probing the complex dielectric functions of the insulator and metallic states (right, adapted from a complex refractive index measurement by Wan et al.[53]). **b** Time-domain HPP near-field interferogram acquired on a VO₂ nanobeam at $T = 0$ ps, with the shifted phase between the ground-state $I_{NF}(t)$ and the photoinduced state $\Delta I_{HPP}(t)$ (inset). **c** The HPP signal amplitude relaxation $R_{HPP}(T)$ measured by the two-phase $T$-dependent (TD) scans at selected locations (top) and the equivalent acquired based on space-domain (SD) scans (bottom). **d** Ultrafast HPP nanoimaging of the VO₂ nanobeam with the AFM topography. **e** The HPP amplitude mapping at $T = 0$ ps, derived from mappings at the two $E_{LO}$ phases, $\phi = 0$ and $\pi$. **f** Time-dependent evolution of the HPP signal along L1 and L2 in panel **e** with the corresponding topography of the nanobeam.

that measurement was performed in SHPP, and the influence of the far-field background on the measured dynamics could not be ruled out[22].

Here, we apply HPP IR s-SNOM to VO₂ nanobeams[51] to quantify the heterogeneous electron dynamics associated with the IMT in a background-free and phase-resolved manner (Fig. 3a). The micro-Raman spectrum of the nanobeams on a silicon substrate shows the VO₂ initially in the M2 phase at room temperature[52] (see Supplementary Fig. S4). Following the NIR pump excitation above the bandgap at 1.2 eV (1030 nm, ~2 mJ/cm² fluence), we measure the HPP interferogram $\Delta I_{HPP}(t)$ with a 0.2 eV (6 μm, 1670 cm⁻¹, ~50 μJ/cm² fluence) mid-infrared probe at $T = 0$ ps (Fig. 3b). We adjust the phase of $E_{LO}$ relative to that of the photoinduced response $\Delta I_{HPP}(t)$ as indicated by $\phi = 0$ and $\pi$ in the inset. Figure 3c (top) then shows the time-resolved HPP amplitude relaxation $R_{HPP}(T)$, acquired at three selected nanobeam locations based on this two-phase measurement.

While at position 2 and 3 the photoinduced infrared response fully relaxes within ~2 ps and signifies excited carrier relaxation, at position 1 the pump-induced signal plateaus beyond 10 ps, characteristic for the formation of the metastable metallic state[10,16,30,33]. By adjusting the pump fluence, we can locally switch the behavior between ps-transient carrier relaxation in the photo-doped insulating state and the photoinduced IMT with its ultrafast nucleation of the metallic state followed by a slower transient domain growth (Supplementary Fig. S4).

Figure 3d then shows the corresponding ultrafast HPP nanoimaging, derived from imaging the sideband-demodulated intensity $\Delta I_{NF}$ for the two $E_{LO}$ phase values $\phi = 0$ and $\pi$ (example shown in Fig. 3e for $T$ ps) as $R_{HPP}(\{X, Y\}) = \Delta I_{NF}(\phi = 0, \{X, Y\}) - \Delta I_{NF}(\phi = \pi, \{X, Y\})$. In addition to the non-uniformity of the decay at individual representative locations as shown in Fig. 3c (bottom), the line profiles across and along the $c_R$ axis of the VO₂ crystal both demonstrate dynamically evolving spatial disorder in the HPP signal amplitude (Fig. 3f).

While the heterogeneity along the $c_R$ axis is particularly pronounced in this example, in other cases a heterogeneity perpendicular to the $c_R$ axis stands out (see Supplementary Fig. S5). By measuring multiple nanobeams with different orientations, we verify that the nonuniform spatial profile of the pump beam does not account for the observed transient heterogeneity, being due to intrinsic heterogeneities in each nanobeam (Supplementary Fig. S6).

As is apparent from Fig. 3b (inset) comparing $I_{NF}(t)$ and $\Delta I_{HPP}(t)$ in time, the transient near-field signal $\Delta E_{NF}$ is phase-shifted from the ground-state response $E_{NF}$, due to the different dielectric response between the insulating and metallic phases as is known for the thermally induced IMT of a VO₂ film (see Fig. 3a, right)[53]. By fitting both the amplitude ($|\Delta E_{NF}|/|E_{NF}| \sim 0.2$) and the optical phase shift ($\Delta\phi = \phi(\Delta E_{NF}) - \phi(E_{NF}) \sim 57°$), we retrieve the actual photoinduced change in the dielectric constant of $\Delta\tilde{\epsilon}_{NF} \sim 0.25 + 1.0i$ at the probe energy of $\overline{v}_{probe} = 1670$ cm⁻¹. In comparison with $\tilde{\epsilon}_{metal}^{therm.} \approx 3 + 84i$ for the thermally induced metallic phase in an extended film sputtered on a silicon substrate[53], our finding of $|\Delta\tilde{\epsilon}_{NF}| << |\tilde{\epsilon}_{metal}|$ implies that the photoinduced excited state is only partially metallic at the pump fluence of ~2 mJ/cm². $Im[\Delta\tilde{\epsilon}_{NF}] > Re[\Delta\tilde{\epsilon}_{NF}]$ is in agreement with the case of a thermally induced metallic phase, with the extracted nano-localized transient dielectric phase $arg(\Delta\tilde{\epsilon}_{NF}) = 76 \pm 2°$ slightly smaller than that of a thermally induced metallic phase $\tilde{\epsilon}_{metal}^{therm.} = 95° \pm 16°$, which is derived from ellipsometry literature values for VO₂ films. With the samples prepared under different conditions in the literature[47,53–56], the deviation might arise from different morphology, strain, or doping, but also a possibly distinct quantum nature of the photoinduced phase in comparison to a thermally induced metallic phase.

A pronounced heterogeneity along the $c_R$ axis (Fig. 3d, f) has been established for thermally induced metallic VO₂ nanobeams and is attributed to nonuniform local strain[10,17,38]. In contrast, the transient heterogeneity in the photoinduced IMT is

predominantly perpendicular with respect to the $c_R$ axis and is consistent with previous work that implied a distinct origin such as intrinsic stoichiometric zoning[16]. The co-existence of the two types of the dynamic heterogeneities both along and across the $c_R$ axis highlights the complexity of the photoinduced IMT, the exact mechanism of which remains unsolved to date. HPP IR s-SNOM, with its space and time resolutions, is thus applicable to address the intracrystalline heterogeneity of a photoinduced IMT in VO$_2$ to guide the development of device applications of VO$_2$ nanostructures with ultrafast control. With quantitative phase resolution, it also lays the groundwork to address the unsolved question regarding the distinct nature of quantum states underlying the photoinduced vs. thermally induced metallic phases[30,32,49].

**Transient vibrational nano-spectroscopy of the polaron–cation coupling in lead halide perovskites**. In the extension of HPP IR s-SNOM to soft and molecular materials, we demonstrate ultrafast vibrational nano-spectroscopy of lead halide perovskites (Fig. 4a) to resolve electron-vibration coupling and its spatial heterogeneity. Lead halide perovskites exhibit an extraordinary optoelectronic response, characterized by the spontaneous formation of long-lived free carriers and long diffusion lengths[57]. Their unusual photovoltaic performance is believed to arise from polaron formation[58], where the charge–phonon coupling extends across multiple unit cells and results in high defect tolerance and coherent carrier transport[2].

As another unique aspect, lead halide perovskites exhibit heterogeneity over multiple length scales in their optoelectronic responses in, e.g., photoluminescence intensity, carrier lifetime, or open circuit voltage[11]. While several works have addressed the nanoscale heterogeneity in the lattice strain and elasticity in the electronic ground-state[59,60], the direct relationship between the nonuniform optoelectronic response and the underlying polaronic heterogeneity has not yet been established. Ultrafast infrared vibrational spectroscopy has previously elucidated the coupling between a molecular cation and a photoinduced polaron in perovskites[35,61] (Fig. 4a), yet was unable to address the underlying spatial heterogeneity due to the diffraction-limited resolution.

Using HPP IR s-SNOM, we aim to resolve the polaron–cation coupling on the nanoscale. To establish the ground and excited-state vibrational responses and their expected coupling to the polaron, we first perform conventional far-field visible (2.4 eV)-pump/infrared (0.2 eV)-probe transmission spectroscopy as shown in Fig. 4b for a thin film of the triple cation

perovskite FAMACs, with chemical composition $[(FA_{0.83}MA_{0.17})_{0.95}Cs_{0.05}]Pb(I_{0.83}Br_{0.17})_3$. We estimate an injected carrier density of ~$10^{19}$ cm$^{-3}$ (for experimental details, see Supplementary Note 4). The pump-induced change in transmission ($-\Delta T/T$) at the pump-probe delay of $T = 0.5$ ps exhibits the transient vibrational signature arising from the CN anti-symmetric stretch mode of the formamidinium (FA) cation[61] in addition to the spectrally broad background from polaron absorption[34]. The vibrational excited-state absorption ($\Delta A_{vib}$) is compared to the ground-state absorption ($A_{vib}$) in Fig. 4b (bottom), exhibiting a blue-shift as well as an absorptive lineshape, suggesting an enhancement in the transition dipole moment. These two observations are consistent with previous observations on similar perovskites[35,61]. As established[35], these two features signify the coupling of the molecular vibration to polaron absorption with a large transition dipole moment and lower resonance energy. This gives rise to the blue-shift and enhanced transition dipole moment of the hybridized polaron-coupled vibration (Fig. 4a). Based on the reported resonance frequency of polaron absorption at 1100–1200 cm$^{-1}$[34,35], the observed vibrational blue-shift of ~5 cm$^{-1}$ corresponds to a spatially averaged polaron–cation coupling strength of ~50 cm$^{-1}$ (see Supplementary Fig. S8 for details).

In ultrafast HPP IR s-SNOM, we then explore the associated nanoscale heterogeneity in polaron–cation coupling (Fig. 5a). Figure 5b shows $\Delta I_{HPP}(t)$ for a pump-probe delay of $T = 2$ ps with the center burst arising from the instantaneous polaron absorption[34] and the long-lived coherence associated with the transient vibrational response. Figure 5c then shows the Fourier transformed spectral profile of the pump-induced $\Delta E_{NF}(\bar{\nu})$ and ground-state $E_{NF}(\bar{\nu})$. The transient vibrational resonant peak at $\bar{\nu}_{ex}$ in Im[$\Delta E_{NF}(\bar{\nu})$] is blue-shifted from the ground-state peak position $\bar{\nu}_{gs}$ in Im[$E_{NF}(\bar{\nu})$] by ~5 cm$^{-1}$, in agreement with the polaron–cation coupling observed in the far-field measurement.

We then apply and fit the data to the model described above (Fig. 2) to retrieve the transient complex dielectric function $\Delta\tilde{\epsilon}_{NF}(\bar{\nu})$. The resonant spectral profile in the retrieved $\Delta\tilde{\epsilon}_{NF}(\bar{\nu})$ (Supplementary Fig. S7) is essentially identical to $\Delta E_{NF}(\bar{\nu})$, with the background carrier response ($\Delta\tilde{\epsilon}_{NF} \sim -0.1 + 0.3i$) qualitatively consistent with polaron absorption. The contribution from Fano-type interference to an apparent shift is negligible in this case compared to the transient vibrational response, due to the relatively small carrier background (Fig. 5d). We thus subtract the carrier background from Im[$\Delta E_{NF}(\bar{\nu})$] to extract the nano-localized excited-state absorption Im[$\Delta E_{NF,vib}(\bar{\nu})$] (see Supplementary Note 4).

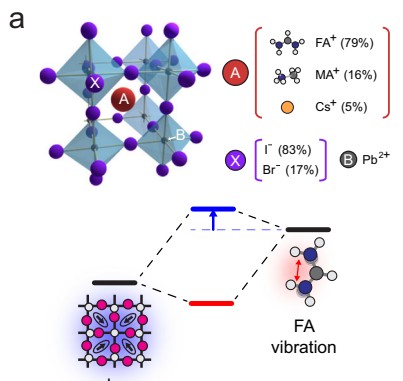
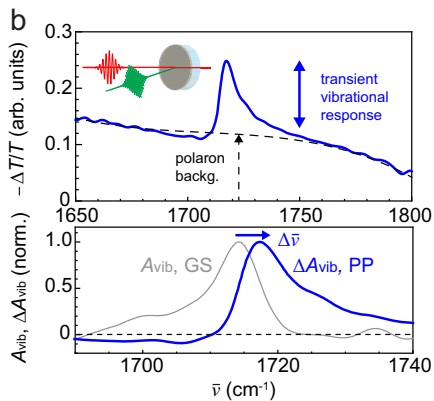

**Fig. 4 Polaron–cation coupling in hybrid organic-inorganic perovskite FAMACs. a** Chemical composition and structure of FAMACs perovskite (top), polaron–cation coupling and associated blue-shift of molecular vibration (bottom). **b** Far-field transmission visible-pump (2.4 eV) IR-probe spectrum at $T = 0.5$ ps (top), the ground-state (GS) vibrational absorbance $A_{vib}$ and transient absorbance $\Delta A_{vib}$ (bottom).

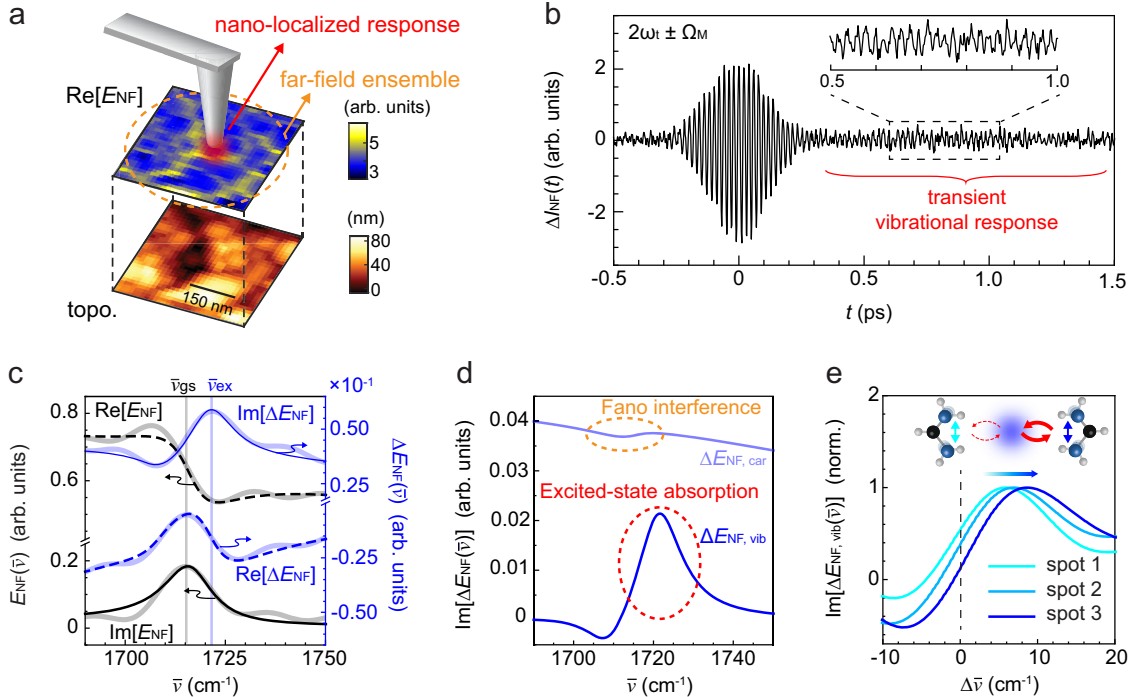

**Fig. 5 Transient vibrational nano-spectroscopy of FAMACs perovskite. a** Ground-state IR s-SNOM Re[$E_{NF}$] imaging (top) and AFM topography (bottom). **b** HPP IR s-SNOM interferogram $\Delta I_{NF}(t)$ acquired at $T = 2$ ps with instantaneous polaron absorption and long-lived transient vibrational coherence. **c** Phase- and frequency-resolved nano-localized pump-probe $\Delta E_{NF}(\bar{\nu})$ and ground-state response $E_{NF}(\bar{\nu})$. **d** Decomposition of Im[$\Delta E_{NF}(\bar{\nu})$] into the transient vibrational (Im[$\Delta E_{NF,vib}(\bar{\nu})$]) and carrier (Im[$\Delta E_{NF,car}(\bar{\nu})$]) contributions with a minor feature from Fano-type interference. **e** The nano-localized transient vibrational signal Im[$\Delta E_{NF,vib}(\bar{\nu})$] at different sample locations shows nanoscale spatial heterogeneity in the polaron–cation coupling.

Figure 5e then shows the nano-localized excited-state absorptions Im[$\Delta E_{NF,vib}(\bar{\nu})$] at three different sample locations. The varying degree of vibrational peak shifts of 5–8 cm$^{-1}$ suggests a spatial heterogeneity in polaron–cation coupling of 50–70 cm$^{-1}$ (Fig. 5e, inset), obscured in the spatially averaged far-field spectroscopy above. Such disordered polaron–cation coupling is likely associated with a non-uniformity in the dynamic lattice elasticity. This interpretation is supported by other recent experimental and theoretical investigations[59,60,62,63], which have identified a heterogeneity in chemical composition and resulting local lattice disorder and strain in perovskite films. With HPP IR s-SNOM probing the excited-state vibrational absorption, we resolve spatial heterogeneities in polaron–cation coupling arising from lattice disorder, which directly impacts polaron formation, lifetime, transport and, as such, photovoltaic device performance.

## Discussion

Recent ultrafast infrared nanoimaging demonstrated the nanoscale probing of a range of low-energy phenomena in semiconductor, 2D, and other quantum materials[16–20,22,24,26], yet mostly with high-repetition excitation that provides sufficient contrast against the simultaneously detected unpumped ground-state response. This conventional approach has therefore been limited primarily to the detection of short-lived nonequilibrium states. This has hampered the application of the technique to access long-lived transient states that often arise from cooperative dynamics associated with many-body interactions[36,37] represented by, e.g., photoinduced phase transitions in correlated electron materials or polaron formation in molecular materials.

In the adaptation of modulated excitation with sideband detection for HPP IR s-SNOM, we facilitate the isolation of the excited-state response from the unperturbed ground-state response, establishing nano-FTIR spectroscopy of the purely transient and nano-localized response with low-repetition excitation. HPP IR s-SNOM thus universally endows ultrafast infrared nanoimaging with the ability to quantitatively resolve ultrafast dynamics associated with long-lived perturbations. We note that HPP s-SNOM, which is compatible with the full range of probe frequencies from visible to far-infrared together with its relatively facile implementation, is complementary to ultrafast EOS s-SNOM probing nanoscale dynamics in the THz regime[29].

As demonstrated in VO$_2$, HPP IR s-SNOM isolates the near-field pump-probe from the ground-state response to achieve ultrafast nanoimaging of the IMT dynamics with high contrast (Fig. 3d). Eliminating the convoluted far-field background contribution by heterodyning $\Delta E_{NF}$ with a phase-controlled $E_{LO}$[22], HPP IR s-SNOM also accurately determines the timescale of the purely nano-localized carrier dynamics (Fig. 1e). Further, by simultaneous and phase-locked recording of $I_{NF}$ and $\Delta I_{NF}$ interferograms, HPP IR s-SNOM quantifies the transient complex dielectric response on the nanoscale (Fig. 3a, b), thus essentially performing ultrafast nano-ellipsometry based on interferometric heterodyne detection. The observed transient spatial hetero-geneity suggests intricate co-existence of competing mechanisms such as nonuniform strain and stoichiometric zoning.

In extensions to transient vibrational nano-spectroscopy, we resolve the generally weak excited-state vibrational coherence of a lead halide perovskite in the time domain (Fig. 5b) with sensitive and selective detection of $\Delta I_{NF}(t)$ enabled by pump modulation. The resulting phase and spectrally resolved excited-state vibra-tional response quantifies the spatially varying polaron–cation coupling (Fig. 5e), which is central to the photovoltaic response of perovskites. Thus the combination of pump modulation and heterodyne detection in HPP IR s-SNOM provides pure transient and nano-localized contrast, extending the applicability of ultrafast infrared nanoimaging from the weakly perturbed to the far-from-equilibrium regime.

More generally, ultrafast nanoimaging based on an electronic excitation and a low-energy probe has a unique advantage in studying the spatial heterogeneity of electron–phonon coupling, a central topic in, e.g., two-dimensional materials[64], hybrid photovoltaics[2], or nanoscale thermal transport[65]. Ultrafast heterodyne infrared nanoimaging, with its direct access to vibrational, phononic, and polaronic modes, can thus play an important role not only in mapping the inherent disorder in electron–phonon coupling[11], but also the optical control of such coupling in combination with nanoscale quantum architectures[23].

HPP IR *s*-SNOM also lays the foundation for adapting other state-of-the-art ultrafast spectroscopy to nanoimaging. For example, by implementing two pump pulses with controlled time delay, ultrafast non-degenerate two-dimensional nano-spectroscopy[4,66] can be realized, probing coherence and population transfer among different modes on the nanoscale. Our implementation with low-repetition excitation is particularly beneficial to potentially implement nonlinear spectroscopy with infrared[67] and THz[68] excitations at the nanoscale, which would require a strong pump fluence that is only attainable in amplifier laser sources. Further, in combination with interferometric heterodyne detection, adiabatic plasmonic nano-focused electronic four-wave mixing[27,28] would provide two-dimensional electronic nano-spectroscopy to characterize local electron or exciton dynamics in, e.g., two-dimensional materials[69].

HPP *s*-SNOM can thus resolve the full spatiotemporal-spectral evolution of key elementary excitations which define the properties of a wide range of functional materials. With the pump modulation and heterodyne detection, HPP *s*-SNOM bridges the prolific success and plethora of modalities of far-field ultrafast spectroscopy to ultrafast nano-spectroscopy and -imaging to probe coupling and dynamics at the nanoscale.

## Data availability

The data generated in this study have been deposited in the Open Science Framework (OSF) at https://osf.io/nkyta/.

## Code availability

The MATLAB codes developed in this study have been deposited in the Open Science Framework (OSF) at https://osf.io/nkyta/.

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

## Acknowledgements

We thank Lei Jin and Junqiao Wu (University of California, Berkeley) for providing VO$_2$ nanobeams, and Jiselle Ye, Prachi Sharma, and Sean E. Shaheen (University of Colorado Boulder) for supplying perovskite samples. J.N., S.C.J., P.T.S.C., D.M.W., and M.B.R. acknowledge support from the NSF Science and Technology Center on Real-Time Functional Imaging (STROBE) under DMR-1548924 for instrument development and the perovskite work. S.A.D., O.K. and M.B.R. further acknowledge funding from the U.S. Department of Energy, Office of Basic Sciences, Division of Material Sciences and Engineering, under Award No. DE-SC0008807 for the work on VO$_2$. J.N. acknowledges support from the Japan Society for the Promotion of Science for a JSPS Overseas Research Fellowship.

## Author contributions

J.N., O.K., S.A.D. and M.B.R. conceived the research. J.N., P.T.S.C., and D.M.W. performed the experiments. J.N., S.C.J., P.T.S.C., D.M.W. and M.B.R. analyzed the data. J.N., S.C.J. and M.B.R. discussed and wrote the manuscript. All the authors read and provided comments on the manuscript.

## Competing interests

The authors declare no competing interests.
