## [Peer Review File · Nature Communications]

Ultrafast infrared nano-imaging of far-from-equilibrium carrier and vibrational dynamicsEditorial Note: This manuscript has been previously reviewed at another journal that is not operating a transparent peer review scheme. This document only contains reviewer comments and rebuttal letters for versions considered at *Nature Communications*. Mentions of the other journal have been redacted.

REVIEWER COMMENTS

Reviewer #1 (Remarks to the Author):

The revised manuscript (currently under consideration in Nature Communications) has been substantially improved over the previous version this referee has reviewed (for {redacted}). I think the authors have adequately addressed the questions and concerns I had in the previous round, and I am happy to recommend its publication in its current form.

Reviewer #2 (Remarks to the Author):

The authors have addressed all of my questions and comments and resolved some of them. They have added due credit to other works and they have made serious attempts to define some of the terms and performance figures. However, some of my most important concerns remain:

1. The main novelty claim needs to be clearer yet.

In the replies the authors clarify that the main novelty claim refers to the combination of s-SNOM with a powerful laser source operating at a relatively low repetition rate of 1 MHz. Such a source allows the authors to observe photoinduced dynamics even in cases where the recovery time takes as long as 1 μ s. I agree that this is a noteworthy success that should be at the center of the novelty claims.

What I do not find helpful is that the authors' discussion remains extremely vague in parts. In the introduction, for instance, they try to prepare the grounds for their results by creating novel categories of scientific disciplines that are not established/not clear/not well defined. Specifically, the authors are creating the impression that so far ultrafast s-SNOM experiments were somehow limited to „weak perturbations“, which were not „far from equilibrium“. I do not understand what this is supposed to say. Obviously, previous pump-probe experiments on graphene and semiconductor nanostructures created extremely non-thermal electronic and structural excitations. Pump fluences of mJ cm^{-2} have been possible. How are such experiments not extremely far from equilibrium? In my view, the authors' attempt to draw an artificial line here does not line up with reality and blurs the actual novelty of the present work. Claims such as the following (page 2) are not substantiated: “Ultrafast infrared nano-

imaging to date has probed ... excitations based on weakly perturbing ... laser pump sources, ...". This artificial distinction even displays in the new title.

The authors are rather recommended to build upon the fact that their low repetition rate allows them to avoid recovery time artefacts and that they reach excellent signal-to-noise ratios. Here, it would be good, however, if they specified their figures of merit in quantitative numbers rather than speculating in qualitative terms.

2. Their extraction algorithm is not new.

In the revised manuscript the authors are giving due credit to relevant previous work, but they still seem to suggest that the dipole model has not been combined with multi-layer description before. Yet this has been done, e.g. in *Opt. Expr.* 20, 13173 (2012), *APL* 107, 151902 (2015), *Nano Lett.* 18, 7515 (2018), *ACS Photon.* 8, 418 (2021).

3. I am still not sure which new information I am supposed to take away from the VO₂ example.

The authors have substantially boiled down their claim that the photo-induced change of the dielectric function at a middle infrared frequency is non-thermal. Nonetheless, they still try to compare the pump-induced and spatially resolved change of the dielectric response with literature values of equilibrium dielectric functions measured in far-field experiments on different VO₂ films. Such literature values can depend on strain, morphology, local material composition, impurities etc. and cannot simply be taken as a quantitative reference to compare. Such comparison would really require measurements on the very same sample.

4. I am still uncertain about what we learn from the perovskite measurements

This discussion requires a number of assumptions. For example, the Fano-like lineshape in Fig. 5D is assumed to be associated with polaron-anion coupling. (Can this not simply be the result of ground state bleaching and excited state absorption?) Anyway, if we accept these assumptions, I do not understand what we learn about the microscopic dynamics here? Anything new?

5. The conclusions do not provide an accurate picture of the novelty of the present work

The authors state: "This approach has therefore been limited to the detection of a strong photoinduced carrier or plasmon polariton response that provides sufficient contrast against the simultaneously detected unpumped ground-state response. This has hampered the application of the technique to access the much wider range of weak responses and interactions ...". The argument is upside down in my view: The authors seem to suggest that the great breakthrough is that they modulate the pump. To the best of my understanding it is extremely simple to modulate the pump (and seems to have been done recently as the authors note in their paper in ref. ...). The only reason why this was not always done, in previous work, apparently is that it was not necessary because of a strong response. Do the

authors really mean to suggest that the main novelty lies in the fact that they modulate the pump? The authors are advised to base their story on a more sustainable argument such as the one that a repetition rate of less than 1 MHz allows them to avoid recovery time artefacts and observe also slow recovery dynamics as explained above.

Based on the present form of the manuscript I would strongly discourage accepting it for publication in Nature Communications, but I would find it fair to give the authors a chance to improve their manuscript such that it accurately reflects its novelty in the context of the existing literature.

General Remark

We thank the two reviewers for their feedback on our revised manuscript. Reviewer #1 now recommends publication without further revisions. Reviewer #2 also acknowledges the novelty of our work in quantitatively probing ultrafast dynamics on the nanoscale as associated with long-lived nonequilibrium states. However, the reviewer requests to further improve the presentation in clarity of the advances of the technique. In addition, Reviewer #2 raises some technical comments which we hope to have now better addressed in the new revision as well.

In our reply, the reviewer comments are reproduced in blue, our reply to the comments are given in black, and then in red we show the corresponding revision in the manuscript.

Reviewer Comments

Reviewer #1 (Remarks to the Author):

The revised manuscript (currently under consideration in Nature Communications) has been substantially improved over the previous version this referee has reviewed (for Nature Photonics). I think the authors have adequately addressed the questions and concerns I had in the previous round, and I am happy to recommend its publication in its current form.

We thank Reviewer 1 for the positive assessment of our revised manuscript.

Reviewer #2 (Remarks to the Author):

The authors have addressed all of my questions and comments and resolved some of them. They have added due credit to other works and they have made serious attempts to define some of the terms and performance figures. However, some of my most important concerns remain:

We thank Reviewer 2 for suggesting how to further improve the manuscript. We revised to provide clarity on the novelty of our work, particularly with the focus on the ability to probe long-lived nonequilibrium states associated with many-body interactions on the nanoscale as we outline below.

1. The main novelty claim needs to be clearer yet.

In the replies the authors clarify that the main novelty claim refers to the combination of s-SNOM with a powerful laser source operating at a relatively low repetition rate of 1 MHz. Such a source allows the authors to observe photoinduced dynamics even in cases where the recovery time takes as long as 1 μ s. I agree that this is a noteworthy success that should be at the center of the novelty claims.

Indeed, implementing ultrafast infrared nano-imaging with low-repetition excitation is a key advance in this work and allows for the study of several important long-lived excitations into the microsecond regime (e.g., photoinduced carriers, phase transitions, polaron formation, thermal dynamics, and others). The photoinduced phase transition dynamics of VO₂ is in fact a typical example and is a model system representing many photoinduced quantum systems that can be

explored only with a sub-MHz amplified laser system [32-35] due to the long-lived transient metallic phase. Similarly, while conventional ~ 100 MHz oscillator sources provide enough fluence for nonlinear spectroscopy with electronic excitation [B. Lomsadze et al., Nat. Photon. 12, 676 (2018) etc.], two-dimensional spectroscopy with infrared [66] and THz [67] excitation has so far only been possible with low-repetition amplified lasers, because of the strong pump fluences requirement. With our demonstration of ultrafast infrared nano-imaging with low-repetition rate excitation, we now pave a way to adapt and generalize these established far-field spectroscopy principles to nanoscale spectroscopy.

We now emphasize more clearly how we make systematic applications of ultrafast s-SNOM with low-repetition excitation possible through the combination of excitation modulation and sideband detection, which facilitates the detection of the weaker pump-probe signal regardless of the reduced duty cycle. For details, please see the specific revisions in response to the questions below.

What I do not find helpful is that the authors' discussion remains extremely vague in parts. In the introduction, for instance, they try to prepare the grounds for their results by creating novel categories of scientific disciplines that are not established/not clear/not well defined. Specifically, the authors are creating the impression that so far ultrafast s-SNOM experiments were somehow limited to „weak perturbations“, which were not „far from equilibrium“. I do not understand what this is supposed to say. Obviously, previous pump-probe experiments on graphene and semiconductor nanostructures created extremely non-thermal electronic and structural excitations. Pump fluences of mJ cm^{-2} have been possible. How are such experiments not extremely far from equilibrium? In my view, the authors' attempt to draw an artificial line here does not line up with reality and blurs the actual novelty of the present work. Claims such as the following (page 2) are not substantiated: “Ultrafast infrared nano-imaging to date has probed ... excitations based on weakly perturbing ... laser pump sources, ...“. This artificial distinction even displays in the new title.

While the reviewer is concerned with the lack of clear quantitative definition of “far-from-equilibrium” or “strongly perturbed” regimes, we indeed follow the established definitions and terminology in the field of ultrafast spectroscopy. Our use of these terms is in accordance with existing literature, yet we acknowledge that additional clarification would benefit the manuscript; for example, the review by Sundaram *et al.*, citation [39] in the revised manuscript, associates the far-out-of-equilibrium state with the modified interatomic forces as a result of a large (10% or more) fraction of the valence electrons excited to the conduction band. Another review by Basov *et al.*, now added as citation [38], roughly categorizes the pump excitation beyond $100 \mu\text{J}/\text{cm}^2$ as a strong perturbation, which leads to a photoinduced phase transition and non-thermal non-equilibrium states. In our work, we in fact explore these scenarios. By these definitions, the photoinduced phase transition in VO_2 and the polaronic carrier excitation in perovskites are in strongly perturbed regimes based on the fluences applied and the electronic excitation that leads to large atomic and molecular displacements, thus resulting in long-lived transient (or metastable) states.

While the exact numbers of fluence and excited electron density are material specific, one of the key features in the far-from equilibrium regime is that cooperative dynamics arising from many-

body interactions, e.g., electron-electron or electron-phonon interactions, lead to the formation of long-lived non-equilibrium states [38,39] as we observe. Our method, with low-repetition excitation and enhanced excited-state contrast, now allows for the systematic investigations of these long-lived non-equilibrium states associated with the strong perturbation on the nanoscale.

To address the reviewer's concern on the uniqueness of the perturbation regimes, we carefully contrast and compare our work with the excitation fluence and dynamic response employed and observed in the literature of ultrafast *s*-SNOM. As we already noted in our manuscript, many pump-probe *s*-SNOM studies have investigated short-lived transients with lower pump fluences (e.g. 30 $\mu\text{J}/\text{cm}^2$ in [31] and 100 $\mu\text{J}/\text{cm}^2$ in [21]). While a fluence as high as 1 mJ/cm^2 was indeed employed in ultrafast EOS *s*-SNOM work on InAs nanowires [26], this still resulted in a short-lived (<1 ps) transient instead of a long-lasting metastable state.

A high fluence of 2 mJ/cm^2 was employed on VO_2 nanowires [19], but the high-repetition excitation (20 MHz) indeed hampered the observation of the dynamics associated with the long lived photoinduced metallic phase. As we demonstrate in our work, we resolve this issue by employing low-repetition excitation with high sensitivity and full heterodyne detection in both the VO_2 and perovskite examples, representing correlated electron and molecular materials, respectively.

Action taken: In the revised manuscript, we clarify that our method enables systematic studies of long-lived ($\leq 1 \mu\text{s}$) non-equilibrium states arising from many-body interactions. We now better contrast our achievement with those in previous studies with their focus on short-lived transients and show how our method provides for a generalized approach with high sensitivity even with low-duty-cycle excitation.

[Abstract (pg. 1)] Ultrafast infrared nano-imaging has demonstrated access to ultrafast carrier dynamics on the nanoscale in semiconductor, correlated-electron, or polaritonic materials. However, mostly limited to short-lived transient states, the contrast obtained has remained insufficient to probe important long-lived excitations, which arise from many-body interactions induced by strong perturbation among carriers, lattice phonons, or molecular vibrations. Here, we demonstrate ultrafast infrared nano-imaging based on excitation modulation and sideband detection to characterize electron and vibration dynamics with nano- to microsecond lifetimes. As an exemplary application to quantum materials, in phase-resolved ultrafast nano-imaging of the photoinduced insulator-to-metal transition of vanadium dioxide, a distinct transient nanodomain behavior is quantified. In another application to lead halide perovskites, transient vibrational nano-FTIR spatially resolves the excited-state polaron-cation coupling underlying the photovoltaic response. These examples show how heterodyne pump-probe nano-spectroscopy with low-repetition excitation extends ultrafast infrared nano-imaging to probe elementary processes in quantum and molecular materials in space and time.

[Introduction (pg. 2)] Important long-lived transients in materials arise from many-body interactions induced by a strong perturbation, such as photoinduced phases in correlated electron systems [32–35] or polaron dynamics in organic-inorganic hybrid photovoltaics [36, 37]. In these systems, cooperative dynamics among the many degrees of freedom [38, 39] result in the formation of non-equilibrium states with nano- to microsecond lifetimes. However, high-repetition excitation precludes quantitative probing of the ultrafast dynamics of those states.

[Introduction (pg. 3)] The induced third-order nano-localized polarization is isolated by sideband lock-in detection and directly detected interferometrically in the time-domain, enabling ultrafast nano-imaging with high contrast even with the low-repetition excitation rate of 1 MHz.

We ensure our use of “far-from-equilibrium” is in accordance with the established use of the term in ultrafast spectroscopy and also clarify in the abstract that this is associated with long-lived transient states induced by strong perturbation, which can now be addressed with our low-repetition excitation and detection scheme. Therefore, we keep the title as it is: “Ultrafast infrared nano-imaging of far-from-equilibrium carrier and vibrational dynamics”.

The authors are rather recommended to build upon the fact that their low repetition rate allows them to avoid recovery time artefacts and that they reach excellent signal-to-noise ratios. Here, it would be good, however, if they specified their figures of merit in quantitative numbers rather than speculating in qualitative terms.

We agree with and appreciate this comment. In the revised manuscript, we quantitatively describe that our method enables the probing of long-lived non-equilibrium states lasting beyond tens of nanoseconds up to 1 μ s, based on our 1 MHz laser excitation. To quantify the enhancement of the signal-to-noise (S/N) ratio introduced by the excitation-modulation and sideband-detection in comparison to the conventional approach based on the detection with unmodulated pump, we now provide the S/N enhancement factor of > 4 (which would reduce the equivalent measurement time by more than one order of magnitude) based on the data given in Figure S9. Together with the ~ 40 nm spatial and ~ 200 fs temporal resolutions, we now communicate all figures of merit more explicitly.

[pg. 5] This detection scheme enhances the signal-to-noise ratio by a factor of > 4 in comparison to the conventional scheme, corresponding to a reduction in the data acquisition time by more than one order of magnitude (Figure S9).

2. Their extraction algorithm is not new.

In the revised manuscript the authors are giving due credit to relevant previous work, but they still seem to suggest that the dipole model has not been combined with multi-layer description before. Yet this has been done, e.g. in *Opt. Expr.* 20, 13173 (2012), *APL* 107, 151902 (2015), *Nano Lett.* 18, 7515 (2018), *ACS Photon.* 8, 418 (2021).

We agree that in previous works the dipole model had already been employed in a multilayer setting for SNOM and we do not intend to claim the combination itself as a novelty in this work. However, while the examples highlighted by the reviewer describe the spectral and phase response in SNOM in the ground state, we extend this scheme to model the transient excited state near-field response. In particular, we model our sideband-demodulated near-field pump-probe signal to be directly comparable to our experimental data. Whereas a similar formalism has been employed by Huber and co-worker in their combination of point-dipole and three-layer models [26], our model, based on finite dipole and four-layer reflection, is more suitable to describe transient vibrational nano-spectroscopy, where the finite dipole model [44] allows to quantitatively retrieve the molecular vibrational response more accurately than the point dipole model with the four-layer model accounting for the limited penetration depth of the excitation pulse into the sample layer.

We also note that, to the best of our knowledge, we, for the first time, apply that theory to describe a transient molecular vibration in nano-spectroscopy. Importantly, we find that in

addition to the excited-state vibrational absorption encoding the molecule-electron couplings, Fano-type interference could also contribute to $\Delta E_{\text{NF}}(\nu)$ as a direct observable in near-field pump-probe spectroscopy. This is a critical insight necessary to accurately interpret the observed resonances not only in transient vibrational near-field spectroscopy (related to the comment 4 by the reviewer), but also for other near-field spectroscopy including, e.g., visible-pump/THz-probe transient phonon nano-spectroscopy.

Action Taken: We now explicitly contrast and compare our model describing the transient response in the excitation-modulated sideband-detection scheme against the previously established combination of dipole model and multilayer reflection model as employed for describing ground-state and ultrafast SNOM with the additional citations [44-46]. We also highlight new insight on the two independent contributions of excited-state absorption and Fano-type interference in transient vibrational nano-spectroscopy, provided by the new combination of finite-dipole model and four-layer reflection model.

[pg. 6] To evaluate excited-state absorption resonances at the nanoscale [20, 23, 25, 26], which contain critical information associated with many-body interactions, we extend the conventional dipole model in combination with multilayer reflection, established for both ground-state [44-46] and ultrafast s-SNOM [20, 26], to quantitatively relate the experimentally observed sideband-demodulated pump-probe interferogram $\Delta I_{\text{HPP}}(t)$ to the underlying transient dielectric function $\Delta E_{\text{NF}}(\nu)$. Importantly, we find that the $\Delta E_{\text{NF}}(\nu)$ as the direct observable in transient vibrational nano-spectroscopy is generally the convolution of the excited-state absorption and Fano-type interference. This underscores the importance of the theoretical framework to retrieve $\Delta E_{\text{NF}}(\nu)$ that purely encodes the excited-state absorption to distinguish these two contributions.

[pg. 7] While the finite dipole model is known to quantitatively retrieve vibrational resonances [44], the four-layer reflection model accounts for partial excitation of a material layer.

[pg. 7] While $\Delta E_{\text{NF}}(\nu)$ exhibits a resonance corresponding to the excited-state absorption at ν_{ex} , it is also compounded by Fano-type interference of the vibrational response with the pump-induced broadband carrier response.

3. I am still not sure which new information I am supposed to take away from the VO₂ example. The authors have substantially boiled down their claim that the photo-induced change of the dielectric function at a middle infrared frequency is non-thermal. Nonetheless, they still try to compare the pump-induced and spatially resolved change of the dielectric response with literature values of equilibrium dielectric functions measured in far-field experiments on different VO₂ films. Such literature values can depend on strain, morphology, local material composition, impurities etc. and cannot simply be taken as a quantitative reference to compare. Such comparison would really require measurements on the very same sample.

We generally agree with this concern. However, we emphasize that the new insights come from the combination of two independent observations, namely i) the spatial nano-imaging of the photoinduced IMT and ii) the quantification of the transient dielectric constant. While the former information is directly relevant for VO₂ device application, the latter is important from a methodology aspect.

Specifically, with spatial nano-imaging, we highlight the distinction of thermal and strain-induced heterogeneity from other contributions (e.g., stoichiometric zoning, defects) that affect the

photoinduced phase transition and do so in an artifact-free manner with heterodyne detection (background free) and low-repetition excitation (recovery-time artifact free). Such information is important in the context of the application of VO₂ nanowires for, e.g., nanoscale ultrafast devices.

For quantification of nano-localized dielectric constant, we fully acknowledge that in an ideal case, the experiment of ground-state ellipsometry and nano-scale pump-probe spectroscopy would be performed on the same sample. However, due to the small sizes of the nanowires and their sparse distributions on the substrate, the application of far-field ellipsometry would be difficult.

The only small difference between our observed transient dielectric phase value of $76 \pm 2^\circ$ in the transient metallic phase in a single micro-crystal and the literature value of $95 \pm 16^\circ$ for VO₂ films prepared on different substrates (silicon and sapphire) with different methods (sputtering and sol-gel) [51] could indeed be from different film morphology, strain, and doping. However, the comparison between these values, although on different systems, is still valuable to show the quantitative phase resolution of our method. It is notable that our measurement captures the expected behavior of the VO₂ metallic phase showing that the dielectric response is dominated by its imaginary part. This lays the methodological basis for a future study of VO₂ using our method to address the outstanding question of the nature of the quantum state underlying the photoinduced transient metallic phase in comparison to the thermally-induced metallic phase.

Action taken: In the revised manuscript, we fully acknowledge the different factors that could contribute to the differences between our and literature dielectric value, other than being a signature of distinct quantum phases. However, we highlight that our work provides the methodological and quantitative basis to address that question.

[pg. 11] $\text{Im}[\Delta\epsilon_{\text{NF}}(\nu)] > \text{Re}[\Delta\epsilon_{\text{NF}}(\nu)]$ is in agreement with the case of a thermally-induced metallic phase, with the extracted nano-localized transient dielectric phase $\arg(\Delta\epsilon_{\text{NF}}(\nu)) = 76 \pm 2^\circ$ slightly smaller than that of a thermally induced metallic phase $\epsilon_{\text{therm, metal}}(\nu) = 95 \pm 16^\circ$, derived from ellipsometry literature values for VO₂ films. With the samples prepared under different conditions in the literature [47, 51–54], the deviation might arise from different morphology, strain, or doping, but also a possibly distinct quantum nature of the photoinduced phase in comparison to a thermally-induced metallic phase.

[pg. 11] HPP IR s-SNOM, with its space and time resolutions, is thus applicable to address the intracrystalline heterogeneity of a photoinduced IMT in VO₂ to guide the development of device applications of VO₂ nano-structures with ultrafast control. With quantitative phase resolution, it also lays the groundwork to address the unsolved question regarding distinct nature of quantum states underlying the photoinduced vs. thermally-induced metallic phases [32, 34, 49].

4. I am still uncertain about what we learn from the perovskite measurements

This discussion requires a number of assumptions. For example, the Fano-like lineshape in Fig. 5D is assumed to be associated with polaron-anion coupling. (Can this not simply be the result of ground state bleaching and excited state absorption?) Anyway, if we accept these assumptions, I do not understand what we learn about the microscopic dynamics here? Anything new?

Foremost, our work shows for the first time that the coupling between molecular cation and photoinduced polaron is spatially heterogeneous. Such coupling has been extensively studied in

recent visible-pump infrared-probe far-field spectroscopy [36, 37, 59, 60] and was reproduced in our own experiment, as shown in Figure 4. Our modeling on the vibrational blue-shift is based on the insights established in these studies. However, the heterogeneity in the polaron-cation coupling could not be resolved due to the lack of spatial resolution. With HPP IR s-SNOM, we successfully resolve such spatial resolution.

The heterogeneity in perovskite films has been an outstanding topic in the field of perovskite optoelectronics, with the mechanism leading to the non-uniformity in, e.g., carrier lifetime or photoluminescence intensity remaining elusive. Our measurement demonstrates the coupling between the photoinduced polaron and cation, which is a part of the perovskite lattice, is disordered. The disorder in the carrier-lattice interaction might provide the missing link to the observed optoelectronic heterogeneity in perovskites.

We also point out that the reviewer appears to misunderstand our interpretation of the data, based on the comment, “the Fano-like lineshape in Fig. 5D is assumed to be associated with polaron-anion coupling”. First, what we address is polaron-*cation* coupling instead of polaron-*anion* coupling, as the molecular vibration that we probe is that of the molecular cation. In addition, and more importantly, the Fano-type interference does not arise from the polaron-cation coupling, and it indeed is the excited-state absorption (together with a relatively minor contribution from ground-state bleach due to the enhancement in the transition dipole moment) that contains the quantitative information on the polaron-cation coupling. In Fig. 5D, we separate the resonance feature observed in $\text{Im}[\Delta E_{\text{NF}}(\nu)]$ into a Fano-type interference (arising from the broad carrier background) and excited-state absorption with ground-state bleach, based on the new modeling described in the theory section. We find that the contribution of Fano-type interference to $\text{Im}[\Delta E_{\text{NF}}(\nu)]$ is minor compared to the vibrational excited-state absorption with the polaron-cation coupling encoded. Therefore, the resonance peak position in the $\text{Im}[\Delta E_{\text{NF}}(\nu)]$ can be regarded as the excited-state absorption peak position, the variation of which would directly reflect the disorder in polaron-cation coupling.

Action taken: In the revised manuscript, we have further clarified that we identify spatial disorder in polaron-cation coupling, which is a new insight that has been inaccessible in former studies due to the lack of spatial resolution. We further modify Fig. 5D to explicitly show that the Fano-type interference is a minor contribution in comparison to excited-state absorption. We also made minor revisions to emphasize it is the excited-state absorption that reflects the polaron-cation coupling through the vibrational blue-shift. See also our revisions to the modeling section on pg.6, where we more clearly communicate the roles of the two possible contributions, excited-state absorption and Fano-type interference.

[pg. 12] Ultrafast infrared vibrational spectroscopy has previously elucidated the coupling between a molecular cation and a photoinduced polaron in perovskites [37, 59, 60] (Figure 4A), yet was unable to address the underlying spatial heterogeneity due to the diffraction-limited resolution.

[pg. 13] The vibrational excited-state absorption (ΔA_{vib}) is compared to the ground-state absorption (A_{vib}) in Figure 4B (bottom),

[pg. 13] We thus subtract the carrier background from $\text{Im}[\Delta E_{\text{NF}}(\nu)]$ to extract the nano-localized excited-state absorption $\text{Im}[\Delta E_{\text{NF, vib}}(\nu)]$ (for further details, see Supporting Information).

[pg. 13] Figure 5E then shows the nano-localized excited-state absorptions $\text{Im}[\Delta E_{\text{NF, vib}}(\nu)]$

[pg. 14] With HPP IR s-SNOM probing the excited-state vibrational absorption, we resolve spatial heterogeneities in polaron-cation coupling arising from lattice disorder, which directly impacts polaron formation, lifetime, and transport and, as such, photovoltaic device performance.

5. The conclusions do not provide an accurate picture of the novelty of the present work. The authors state: “This approach has therefore been limited to the detection of a strong photoinduced carrier or plasmon polariton response that provides sufficient contrast against the simultaneously detected unpumped ground-state response. This has hampered the application of the technique to access the much wider range of weak responses and interactions ...”. The argument is upside down in my view: The authors seem to suggest that the great breakthrough is that they modulate the pump. To the best of my understanding it is extremely simple to modulate the pump (and seems to have been done recently as the authors note in their paper in ref. ...). The only reason why this was not always done, in previous work, apparently is that it was not necessary because of a strong response. Do the authors really mean to suggest that the main novelty lies in the fact that they modulate the pump?

Indeed, ultrafast infrared nano-imaging with low-repetition excitation to access long-lived excited state dynamics became only possible with the combination of excitation modulation and sideband detection scheme. While adding pump pulse modulation is in principle straightforward with the introduction of a mechanical chopper or AOM, it is still non-trivial given the overall complexity and far from routine combination of s-SNOM with ultrafast spectroscopy. Our work presents, to the best of our knowledge, the first example of near-field pump-probe “synchronized parallel” detection, where the pump-probe signal is demodulated at the sum and difference frequencies between the tip harmonics and pump modulation frequencies. The resulting excited state contrast, which is free from the offset from the ground-state response, is a key enabling achievement to perform both two-phase pump-probe relaxation measurements (Fig. 1E) and imaging (Fig. 3D) and transient vibrational nano-spectroscopy with high sensitivity (Fig. 5B). In the conclusion we hope to convey this technical advance that facilitates these explorations of new regimes.

The authors are advised to base their story on a more sustainable argument such as the one that a repetition rate of less than 1 MHz allows them to avoid recovery time artefacts and observe also slow recovery dynamics as explained above.

We agree with the reviewer that we should emphasize the central advance of the low-repetition excitation in the conclusion section.

Action Taken: We revised the conclusion accordingly now better emphasizing the central advance of low-repetition excitation, which enables probing of ultrafast dynamics of long-lived non-equilibrium states, and also the combination of excitation modulation and sideband detection that facilitates the detection of the near-field pump-probe signal even under that low duty cycle.

[Conclusion pg. 15] This conventional approach has therefore been limited primarily to the detection of short-lived non-equilibrium states. This has hampered the application of the technique to access long-lived transient states that often arise from cooperative dynamics associated with many-body interactions [38, 39], represented by, e.g., photoinduced phase transitions in correlated electron materials or polaron formation in molecular materials.

[Conclusion pg. 15] In the adaptation of modulated excitation with sideband detection for HPP IR s-SNOM, we facilitate the isolation of the excited-state response from the unperturbed ground-state response, establishing nano-FTIR spectroscopy of the purely transient and nano-localized response with low-repetition excitation. HPP IR s-SNOM thus universally endows ultrafast infrared nano-imaging with the ability to quantitatively resolve ultrafast dynamics associated with long-lived perturbations.

[Conclusion pg. 16] Our implementation with low-repetition excitation is particularly beneficial to potentially implement nonlinear spectroscopy with infrared [66] and THz [67] excitations at the nanoscale, which would require a strong pump fluence that is only attainable in amplifier laser sources.

Based on the present form of the manuscript I would strongly discourage accepting it for publication in Nature Communications, but I would find it fair to give the authors a chance to improve their manuscript such that it accurately reflects its novelty in the context of the existing literature.

With the comments and revisions above we hope to have addressed the reviewer's remaining concerns by clarifying that the central advance in our method is the low-repetition excitation, which enables the quantitative evaluation of ultrafast dynamics associated with long-lived (≤ 1 μ s) non-equilibrium states arising from strongly perturbed many-body interactions (for which we now also provide a better physical definition). We also emphasize that the detection of the small signal associated with the low-duty cycle is facilitated with the excitation modulation and sideband detection scheme.

For the modelling section, we emphasize that our approach directly models a transient response in sideband detection. Its implementation with finite dipole and four-layer models allows for the quantitative separation of the observed resonant feature in $\text{Im}[\Delta E_{\text{NF}}(\nu)]$ into a Fano-type interference (arising from the broad carrier background) and excited-state absorption in transient vibrational nano-spectroscopy.

In the example of VO_2 , we provide for unambiguous evidence of intra-crystalline spatial heterogeneity in the photoinduced metallic phase that would directly impact nanowire-based devices. In the application to perovskite, we identify the disorder in polaron-cation coupling based on the blue-shifts in the excited-state absorption.

The revised manuscript now accurately reflects the standing and advance of our work in relation to the existing literature of ultrafast near-field spectroscopy.